# Cognitive Defusion and Psychological Flexibility Predict Negative Body Image in the Chinese College Students: Evidence from Acceptance and Commitment Therapy

**DOI:** 10.3390/ijerph192416519

**Published:** 2022-12-08

**Authors:** Shuanghu Fang, Dongyan Ding, Pingping Ji, Mingjie Huang, Kesong Hu

**Affiliations:** 1School of Educational Science, Anhui Normal University, Wuhu 241000, China; 2Department of Psychology, Lake Superior State University, Sault Ste. Marie, MI 49783, USA

**Keywords:** acceptance and commitment therapy (ACT), cognitive defusion, psychological flexibility, negative body image, Chinese college students

## Abstract

Body dissatisfaction is a global phenomenon. Despite the significant cultural difference, most research on negative body image was conducted in Western countries. How do cognitive fusion and psychological flexibility relate to negative body image in the Chinese population? In the present study, this question was investigated through the intervention technique, Acceptance and Commitment Therapy (ACT). Here, 86 young Chinese university students with high negative physical self were invited, in which 42 students received 10 sessions of group-based ACT intervention in a clinical setting while the remained acted as the control group with no intervention. Pretests showed no statistical differences in negative body image between these two groups, while both cognitive fusion and psychological flexibility predicted negative body image. Post-pre tests showed no change in the control group, while enhanced cognitive defusion and psychological flexibility in the ACT group. Individual differences in psychological flexibility and cognitive defusion enhancement predicted improved body image. A strong association of implicit body image with Fatness and Shortness changes suggested that although with individual differences, those components could be internalized during the intervention in the College students.

## 1. Introduction

Have you ever looked in the mirror and reflected “If I could lose five pounds, how happy I would be?” Body appearance is one of the most important aspects of one’s self-image. As a multi-dimensional construct, body image refers to the mental representation of one’s own physical characteristics, such as weight, shape, height, and skin color [1,2,3]. From the psychological perspective, body image includes perceptual (e.g., body size estimation), cognitive (thoughts and beliefs about the body), affective (body feelings), and behavioral components (e.g., body checking) [1,2,3]. Positive body image is surely the true perception of our body, seeing the parts of our body as they are. Unfortunately, the majority of the population in both Western and Eastern cultures are dissatisfied with their body image [4], and among them, most take negative thoughts, perceptions, and attitudes to their physical appearance. For instance, in a study with 9667 Western women, researchers found that many demonstrated weight-based body dissatisfaction, with the majority (>84%) wanting to be thinner [5]. In another adult study, almost 60% of women and 40% of men had body dissatisfaction, and these rates stayed stable across the lifespan [6]. Even with preadolescents, approximately 40% of boys dislike their bodies [1,7]. Body dissatisfaction and unfavorable social comparisons represent nuclear risk factors for psychological distress [8,9] and mental disorders, including cognitive biases [10] and eating disorder [11].

It is well known that individuals’ attitude toward their body image is rooted in their social cognition and culture [12,13,14,15]. While western culture emphasizes individual responsibility in different life domains, including self-management and weight control [16], this is less highlighted in the Eastern cultures. In China, traditional Chinese culture emphasizes inner traits far more than outward body image when seeking a potential spouse [17]. People’s attitude to body image is specifically influenced by wealth, social hierarchy, not simply by the outward appearance in China [18]. As an important index of the physical self, thinness typically indicates a female’s beauty, good health, self-discipline and sexual attractiveness in Western societies [19], while a sign of poverty and malnutrition in Eastern countries, including China, Korea, and Philippines [20]. Although recent economic and cultural modernizations have shifted the landscape to more or less to Western culture [14,18,21], the traditional culture and its influence is long-lasting [22]. Culture may influence negative body image thinking on two aspects, for instance, body dissatisfaction thoughts and the way that thoughts occur. Despite the significant cultural differences [23,24,25], most research for the negative body image are investigated in Western countries [5,6].

It has been hypothesized that psychological flexibility is cornerstones of psychopathology [26]. Psychological flexibility is a key ingredient in psychological health. As an integrative process, psychological flexibility refers to the process of contacting the present moment fully as a conscious human and persisting or changing behavior in the service of chosen values [27,28]. Psychological flexibility includes several interacting processes, such as acceptance, cognitive defusion, self-as context, openness, values, and committed action [28]. In the contemporary therapy, psychological flexibility incorporates processes of operant conditioning and a specific conceptualization of cognition and cognitive influence [29]. Exactly, in recent decades, psychological flexibility has a multitude of names, including ego-resiliency, executive control, response modulation, and self-regulation [30]. It has been documented that interventions that focus on psychological inflexibility could be useful to treat symptoms associated with eating disorders, overweight, obesity, axiety, body shape dissatisfaction, and weight self-stigma and thus decrease the negative body image [31,32,33,34,35,36,37,38].

Cognitive fusion, the opposite side of cognitive defusion, involves attaching a thought to an experience [39]. When one’s mind tells them how to balance their bank account or drive car safely, listening to the mind could be adaptive. However, it is different if their mind says that they are boring or unattractive [39,40]. When fused with their thoughts, people tend to respond to them as if they were facts, or they represent the truth, triggering experimental avoidance strategies, and making these internal experiences more painful. The previous research argued that cognitive fusion has a negative impact on body image and unhelpful body image coping strategies [11,41].

Several unique systems of intervention have emerged in recent decades, such as Acceptance and Commitment Therapy (ACT). ACT aims to improve individual psychological flexibility [28,42], and cognitive defusion has been identified as a mediator of treatment outcomes in ACT [43]. Not just ACT, some have suggested that cognitive defusion is an important general supplement to several clinical approaches and techniques to change one’s relationship to or functions of thoughts and feelings [44]. Previous studies in western countries have suggested that cognitive defusion may have an important influence on eating disorder symptom severity and body image [41,45]. If this is general across cultures, we expect that cognitive fusion relates to negative body image and its improvement, respectively. If it is special, cognitive intervention may have no effect or influence, only special components reflected in negative body image construct.

Although cognition defusion has a close relationship with psychological flexibility, research on this topic has been fragmented, with a few attempts at synthesis and interpretation [26,46,47]. It remains unclear how cognitive fusion and psychological flexibility relate to negative body image in the Chinese population. In the present study, we evaluated whether individual differences in cognition fusion/defusion predict negative body image improvement and the relationship between cognitive fusion and psychological flexibility in the Chinese population.

The overall aim of this study was to examine the relationship between cognitive fusion/defusion, psychological flexibility, and negative body image in Chinese university students. Could cognitive changes in negative body image be internalized during the cognitive defusion intervention? There are several intervention methods we can consider [48,49]. Our method took advantage of the ACT approach [50], as it specifically includes the cognitive defusion intervention, with special sessions to improve participants’ psychological flexibility (see below). We used standard scales for cognitive fusion/defusion, psychological flexibility, and negative body image assessment. By doing so, we also explored if the ACT is effective in intervening with negative body image. In summary, the main purpose of this study is to explore whether group-based acceptance and commitment therapy can reduce negative body image among college students and what are the intervention mechanisms involved.

## 2. Method

### 2.1. Participants and Design

The study was conducted at the first author’s university. Using a convenience sampling method, we contacted university counselors and distributed 1000 questionnaires. 968 questionnaires were returned (96.8% effective rate). We identified 463 of the 968 surveys with an average score on each dimension of negative body image greater than 3. Among them, there were 244 males and 219 females, aged between 17 and 22. Participants in this study were recruited from these 463 university students. The inclusion criteria of intervention group were being aged 18 years or older and self-reporting average score on each dimension of negative body image greater than 3. Exclusion criteria assessed via self-report measures and one-to-one online semi-structured interviews with a trained research assistant were suicidal intentions, use of psychiatric medicines, receiving psychological counseling or other treatment, and not being willing to undergo the ACT intervention. 46 participants, including 22 males and 24 females, were recruited as the intervention group. Four participants withdrew from intervention due to their private reasons in the later stage of the process. There were 42 participants in the intervention group, including 19 males and 23 females, aged from 18 to 22. At the same time, 46 out of the remaining students were randomly selected as the control group (2 participants withdrew for their private reasons), Finally, the control group consisted of 44 participants, including 21 males and 23 females, aged from 18 to 22, during which no intervention was performed, See Table 1 for details. The statistical power of the sample size was calculated using G*Power 3.1 [51]. The statistical power for a sample size of 86 is 0.96 when ANCOVA was used with group size of four, effect size (f) of 0.4, and a significance level of 0.05 [52].

The study was 2 (condition: intervention vs. waitlist control) × 2 (time: pre- and post-test) mixed design, with a final sample of 86 students (ACT intervention group: *n* = 42; waitlist control group: *n* = 44). G*Power 3.1 showed that the statistical power is 0.94 when linear regression was used with a sample size of 86, default effect size (R^2^) of 0.15, and a significant level of 0.05.

This study was approved by the Ethical Committee of Anhui Normal University (approval number: 2018076). All students voluntarily participated in the research. All participants were treated under the Declaration of Helsinki and its latest amendments and provided written informed consent before participating in the study.

### 2.2. ACT Procedure

After the pretest, the participants in the intervention group participated in ACT workshops based on the six modules of ACT [28], while students in the waitlist control group received no interventions. The intervention protocols were based on the translated version of the “Act in practice: Case Conceptualization in Acceptance and Commitment Therapy” [53] and were divided into ten sessions. The sessions focused on the theme of “Face the negative body image and do your best,” which lasted for 5 weeks, twice a week, and about 1.5 h each time. The intervention protocols were included in Table 2. The intervention protocols were evaluated and approved by an Acceptance Commitment Therapist, and the entire intervention was supervised by an Acceptance Commitment Therapist. Participants in the control group received normal college education, hence no intervention.

### 2.3. Measures

#### 2.3.1. Psychological Flexibility

We measured participants’ psychological flexibility using the Chinese version of the Acceptance and Action Questionnaire II (AAQ-II) [54]. The previous study has shown this scale has excellent reliability and validity [55]. This questionnaire includes seven items (e.g., “I’m afraid of my feelings”) that are rated on a 7-point Likert scale (1 = *never*, 7 = *always*). Higher AAQ-II scores indicate lower levels of psychological flexibility. In the current study, internal consistency in this sample was excellent (α = 0.91).

#### 2.3.2. Cognitive Fusion

The Chinese version of the Cognitive Fusion Questionnaire (CFQ) was used [56]. We used the cognitive fusion questionnaire (CFQ-F). This questionnaire included 9 items (e.g., “My thoughts cause me distress or emotional pain”) that were rated on a 7-point Likert scale (1 = *never*, 7 = *always*). The higher the score, the higher the degree of cognitive fusion. In the present study, Cronbach’s α was 0.93.

#### 2.3.3. Negative Body Image

We used the Negative Physical Self Scale (NPSS) to measure participants’ negative body image [57]. It has been reported that NPSS has more excellent utility for assessing body image disturbances in China than existing measures that focus exclusively on general body satisfaction and body size/weight as Chinese adolescents and young adults express relatively more concerns about General Appearance, Shortness, and Facial Appearance than about Fatness. As a multidimensional measure of body image, NPSS has 48 items consisted of five dimensions (General Appearance, Facial Appearance, Shortness, Fatness, Thinness) and is rated on a 5-point Likert scale (1 = never, 5 = always). The scale has excellent test–retest reliability and criterion validity. In the present study, Cronbach’s α is high for General Appearance (0.80), Facial Appearance (0.86), Shortness (0.92), Fatness (0.90), and Thinness (0.79).

#### 2.3.4. Implicit Association Test (IAT)

We wanted to see whether, and if so, how the ACT would modify the implicit attitudes to the negative body image attitudes in the Chinese population. In the present study, IAT was used to measure the unconscious bias to the negative body image and its change. The program was created on E-prime version 2.0. This test was conducted in a quiet assessment lab.

Consistent with the standard IAT [58,59], here, the test had ten, respectively, self and non-self conceptual words, and positive and negative body image words (all presented in Chinese). A series of steps were processed to develop and validate the word stimuli used for the IAT here. First, based on the literature, 20 positive body self-words and 20 negative body self-words were identified [60,61]; Second, 80 undergraduate students not included in the main study were recruited to rate the words’ valence on a 7-point scale (“most negative” to “least positive,” 1 to 7) and judge whether these words describe the negative body image; and finally, ten words with the highest valence were selected as the target words. The compatibility tasks combined self-related words and positive adjectives, or the combination of nonself-related words and negative adjectives. The incompatible tasks combined self-related words and negative adjectives, or a combination of nonself-related words and positive adjectives. Participants differently reacted to the combinations of two kinds of words. The tests were divided into 7 steps, as shown in Table 3. Each stage was preceded by a set of instructions about the dimensions of the categorization task and the appropriate key response with Q key and P key.

Consistent with the literature [59,62], three main steps were adopted to extract the indexes of the implicit body image: First, the data from the first two responses in the compatible task and incompatible task were deleted because participants were easily affected by other interfering factors at the beginning of the test; Second, the reaction time longer than 3000 ms was calculated as 3000 ms, and the reaction time shorter than 300 ms was calculated as 300 ms; and third, the data with an error rate of over 20% were eliminated, and the data of the 4th and 7th trial blocks were adopted as the response time of compatible tasks and incompatible tasks. The mean value of the compatible task response time and the incompatible task response time of each participant were taken, and the logarithmic transformation of which was carried out. The difference between the mean logarithm of the two was chosen as the index of implicit body image. All participants were tested simultaneously before and after the treatment (i.e., pre- and post-test).

### 2.4. Analysis

The significance level for the main statistical analyses was set at *p* < 0.05. In the present study, effect sizes were reported as *d* (small = 0.30; medium = 0.50; large = 0.80) for planned comparison *t*-tests and one-way ANCOVA [63,64]. We used Pearson correlation and linear regression analysis to explore potential relationships between measures (effect size was reported as R^2^).

To test the models of cognitive defusion and psychological flexibility relate to negative body image, we performed a mediation analysis [65]. A standard mediation model involves evaluating these four components: (1) total effect *C* (initial variable → outcome, which equals to the direct effect *c’* plus the indirect effect *a*b*); (2) indirect path *a* (initial variable → intervening variable, mediator; (3) indirect path *b* (intervening variable → outcome, after controlling for the initial variable; and (4) direct effect *c’* (initial variable → outcome, after controlling for the intervening variable). The final mediation effect is tested by assessing the significance of the product of paths a and b by using bootstrap tests [66,67]. For recent applications and discussion, see Hu, K. [68,69].

## 3. Results

### 3.1. Pre-Test

To establish a baseline, we first confirmed there was no difference between ACT and control groups on critical measurements before the intervention. As expected, the pretest demonstrated the variance of all data were homogeneous, and there were no significant differences between ACT group and control group on cognitive fusion (CFQ), psychological flexibility (AAQ) and negative body image indexes (NPSS), largest t(84) = 1.44, *p* = *0*.154, Cohen’s *d* = 0.31 (Figure 1, gray bars). Further analysis showed no differences on the five components involved in negative body image (i.e., General Appearance, Facial Appearance, Shortness, Fatness, Thinness) between ACT and control groups, largest t(84) = 1.29, *p* = *0*.200, Cohen’s *d* = 0.33.

### 3.2. Post- vs. Pre-Test

We expect that no effect would be observed in the control group, and this was supported exactly by the post- vs. pre-tests on cognitive fusion, psychological flexibility and negative body image, largest t(43) =1.07, *p* = 0.292, Cohen’s *d* = 0.03 (Figure 1, see control group side). Compared to that no change was observed in the control group, post-intervention significantly improvements were obtained in the ACT group: Cognitive fusion was decreased, t(41) = −3.05, *p* = 0.004, Cohen’s *d* = 0.54; psychological inflexibility was decreased, t(41) = −2.27, *p* = 0.029, Cohen’s *d* =0.48; and negative body image attitude was decreased, t(41) = −8.91, *p* < 0.001, Cohen’s *d* = 1.03. As negative body image attitude was significantly improved, we further scrutinized the five components involved in NPSS (i.e., General Appearance, Facial Appearance, Shortness, Fatness, and Thinness). It showed that all components were improved, smallest t(41) = 4.07, *p* < 0.001, Cohen’s *d* = 0.39.

### 3.3. Post-Test

To inform the differences between two groups in the post-test, we ran ANCOVA in which the pre-measurement was a covariate. The interaction terms between the independent variable and covariance were not significant. The results showed significant differences between two groups on psychological inflexibility (*F*(1, 83) = 4.18, *p* = 0.043, *d* = 0.32) and cognitive fusion (*F*(1, 83) = 7.67, *p* = 0.07, *d* = 0.43). The five components involved in NPSS were also significantly different between the two groups, which indicated the ACT group had higher scores, the smallest *F*(1, 83) = 16.45, *p* < 0.001, *d* = 0.63. Table 4 presents the descriptive statistics and difference tests between the ACT intervention group and the control group, along with the effect sizes.

### 3.4. Cognitive Defusion during Intervention

Is psychological flexibility enhancement the underlying basis for negative body image improvement? What is the role of cognitive defusion then? Further analyses revealed a significant correlation between cognitive fusion (post) and negative body image scores (post), r = 0.464, *p* = 0.002. Psychological flexibility (post) and cognitive defusion have (post) the strongest correlation, *r* = 0.568, *p* < 0.001, Significant positive correlation between psychological flexibility (post) and negative body image (post), *r* = 0.389, *p* = 0.011.

To further examine the role of cognitive fusion in negative body image improvement, we ran a mediation analysis with three variables: Psychological Flexibility_Post-Pre_ (IV), Cognition confusion/CFQ_Post-Pre_ (mediator), and NPSS_Post-Pre_ (DV). A bias-corrected 95% bootstrap confidence interval (CI) for the indirect effect (a*b = 0.2039) based on 10,000 bootstrap samples was entirely above zero (0.0290~0.4490), indicating a statistically significant mediation model. As seen in Figure 2, the psychological flexibility influence on negative body image attitude was mediated by cognitive defusion. These results highlighted the importance of cognitive confusion on negative body image improvement. To confirm this is the mechanism for the ACT on the negative body image improvement, we also tried a second model with only the pre-test data (*n* = 86). Despite that, the three factors (i.e., Psychological Flexibility, Cognition confusion, and NPSS) correlated with each other, a bias-corrected 95% bootstrap confidence interval (CI) for the indirect effect (a*b = 0.0072) based on 10,000 bootstrap samples showed a nonsignificant mediation model (−0.1485~0.3245).

### 3.5. Individual Differences in Implicit Processing

Our Implicit Association Test (IAT) afforded an examination of whether the cognitive intervention (i.e., ACT training here) influenced implicit attitude change on negative body image. No intervention effect was observed on IAT in the ACT group (t(39) = 1.47 *p* = 0.150, Cohen’s *d* = 0.32), or the control group (t(39) = 0.51, *p* = 0.616, Cohen’s *d* = 0.52). Not surprisingly, IAT change (i.e., Post-Pre) was significantly positively correlated with the negative body image enhancement, *r* = −0.518, *p* = 0.001. Whether the IAT effect may also reflect specific components involved in negative body image (i.e., General Appearance, Facial Appearance, Shortness, Fatness, or Thinness)? See Figure 3, Further analysis showed that IAT change predicted Fatness attitude improvement, *r* = −0.354, *p* = 0.025, and shortness effect (post-pre), *r* = −0.379, *p* = 0.016. Then, we used linear regression analysis to examine the predictive effects of IAT changes on Fatness and and shortness attitude improvement, respectively. The results showed that after controlling for the effects of gender and age on negative body image, IAT changes significantly predicted fatness attitude(*β* = −0.39, *p* = 0.03) and shortness attitude (*β* = −0.36, *p* = 0.016), The above results showed a unique role of implicit attitude in negative body image. See Table 5.

We further tested the correlation between the Fatness effect (post-pre) and Shortness effect (post-pre), *r* = 0.108, *p* = 0.506. The nonsignificant association suggested that ACT intervention possibly modulated the two components (i.e., Fatness, Shortness) differentially. A smaller IAT effect means that the cognitive intervention decreased the unconscious negative words, especially the self-relevant processing. These related to the question if the mechanism by which cognitive defusion and cognitive restructuring techniques have a long-term impact on negative body image change.

## 4. Discussion

In the present study, we investigated the relationship between cognitive defusion, psychological flexibility, and negative body image in Chinese College students. We specifically took advantage of the ACT approach as it consists of cognitive defusion meditation practice, with widely-used special sessions that improve participants’ psychological flexibility in Western culture. We showed that (1) cognitive fusion correlated with psychological flexibility, and each well predicted body dissatisfaction in the higher negative body image sample; (2) Cognitive defusion predicted negative body image improvement. Psychological flexibility was enhanced via ACT intervention, yet its influence on body image attitude was mediated by cognitive defusion; (3) ACT intervention modulated the implicit processing of fatness and thinness, which are special in Chinese culture. In the following, we discuss our findings at greater length.

The majority of the population is dissatisfied with their own body image [5,6]. Although negative body image thoughts may be normative experiences, how one responds to is crucial as it could be associated with a range of mental health issues, including eating disorders and depression [70]. For persons with body image concerns, biased information processing could be influenced and reflected on several aspects, such as excessive attention towards perceived flaws in one’s body, recall chiefly about negative past experiences about one’s body, and fatness-based interpretations of ambiguous stimuli [21,71,72,73,74]. Further, culture may modulate negative body image thinking, for instance, body dissatisfaction thoughts and how thoughts occur [15]. There are several intervention methods developed to intervene with negative anxiety disorder and negative body image; however, so far, most research was conducted in Western culture [19,26,48]. Rather than discussing the difference between intervention techniques, we focus on two primary components among all interventions, i.e., cognitive defusion and psychological flexibility, which is highlighted exactly in the ACT [30,75]. Our pre-test data demonstrated that they both predicted negative body image thinking in the Chinese population. In detail, the more serious cognitive fusion, the more serious with negative body image, and the more serious with psychological inflexibility, the more serious with negative body image. Furthermore, cognitive fusion was associated with psychological inflexibility. These data indicated that cognitive fusion and psychological inflexibility are perhaps two cornerstones of negative body image in the Chinese population.

Cognition fusion is known as a tendency for behavior to be overly regulated and influenced by cognition. Put another way; cognitive fusion pulls us away from living in alignment with our values. Psychological inflexibility, on the other hand, refers to a “rigid dominance of psychological reactions over chosen values and contingencies in guiding actions” [76], which often occurs when people attempt to avoid experiencing private events. In this view, acceptance and experiential avoidance are seen as examples of psychological flexibility and inflexibility. Previous studies have also suggested avoidance as an important factor influencing body dissatisfaction [77], while self-acceptance consistently emerged as a protective factor relative to all negative body images [78]. At the same time, cognitive fusion is related to the distress of avoidance behavior and other unfavorable outcomes across a wide range of physical and mental disorders [79]. Previous studies have shown that ACT interventions are beneficial for participants with weight self-stigma, eating disorders, shame, weight-related experiential avoidance, self-criticism, and body mass index [32,80,81,82]. Our research went beyond these findings by showing that ACT interventions can decrease cognitive fusion and enhances psychological flexibility, and in this way, decrease the negative body image in the Chinese sample. First, ACT intervention decreased cognitive fusion, increasing cognitive defusion. The more increased cognitive defusion, the lower the negative body image. Second, ACT intervention decreased psychological inflexibility, increasing psychological flexibility. Although psychological flexibility improvement was positively correlated with cognitive defusion, psychological flexibility improvement was not correlated with negative body image improvement. This is important, as this pattern differs from what we reported above (i.e., pre-test). The influence of psychological flexibility enhancement on negative body image was mediated by cognitive defusion. These data together showed that cognitive defusion, one of the most important goals with ACT, played a vital role in negative body image improvement. Understanding how cognitive fusion affects mental health is a relatively new area of research. In this study, it is a kind of cognitive defusion intervention. With other therapy CBT, researchers also reported body image improvement significantly [83,84]. Cast in broader terms, although we employed ACT in this study, any therapy that emphasizes the defusion or restructuring of negative thoughts from reality might be helpful in the negative body image intervention [46,48,83,84,85].

It is generally accepted that an individual’s attitude toward his/her body image is rooted in social cognition and culture [12,13,14]. China is largely influenced by communist ideology, Confucian, and later Neo-Confucian orientation [86]. In the meanwhile, people’s attitude to body image is specifically influenced by wealth, and social hierarchy, not simply by outward appearance in China [18]. With a cultural system different from that of western countries, this country has the planet’s most rapidly expanding economy and 20% of its population, whose social transformation has a great impact on the public psyche [57]. Our correlation findings indicate that ACT intervention had effects on the implicit processing of Fatness and Shortness, although with large individual differences. Further, the Fatness and Shortness changes did not correlate with each other. We found no correlation of implicit processing with other components, such as General appearance, Facial appearance, and Thinness. The two components, fatness, and shortness are special, as they are likely responsible for appearance-fixing behaviors and experimental avoidance in Chinese culture. Together, these findings revealed that ACT intervention changed participants’ negative body image even under the control of automatically activated evaluation, without the performer’s awareness of that causation. This opens a new question if the mechanism by which cognitive defusion and cognitive restructuring have a long-term impact on negative body image change. Our initial data partly supported this opinion. The present data could not let us conclude if this is special with the ACT or general with any therapy in which cognitive defusion and psychological flexibility improvement are involved.

The present study measured psychological flexibility using the Acceptance and Action Questionnaire II (AAQ-II), which measures only one dimension of psychological flexibility. In addition, this study did not have follow-up measurements to examine the durability of the intervention study effects. Given these limitations, the findings from this study should be interpreted with caution. Although theorists separate cognitive fusion/defusion and psychological flexibility, cognitive fusion is an important process central to or outcome of psychological inflexibility. The present study was not planned to test the difference of each component that psychological flexibility may consist of, nor the specific construct of cognitive defusion and psychological flexibility. Rather than emphasizing the relationship, we highlighted the role of cognitive defusion in negative body image regulation. By doing so, we provided evidence that group-based ACT intervention could improve university students’ negative body image. Here, all are based on the self-report and observed behavioral changes. Because of the theoretical importance of body image in current psychological models, further research is needed to reveal neural mechanisms associated with the cognitive representative of body image pre- and post-interventions, both in cultural difference and self-referential processing. Follow-up research is needed to test whether other formats of interventions (e.g., home-based or self-help intervention) have similar effects on negative body image with group-based interventions.

## 5. Conclusions

The present study found that ACT intervention can enhance cognitive defusion and psychological flexibility for college students with a high negative physical self. Besides, individual differences in psychological flexibility and cognitive defusion enhancement can predict improved body image. In addition, the strong association of implicit body image with Fatness and Shortness changes suggests that cognitive fusion and psychological flexibility were internalized during the intervention.

## Figures and Tables

**Figure 1 ijerph-19-16519-f001:**
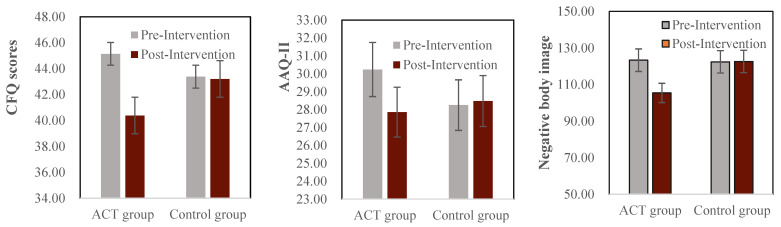
Bar plots for cognitive fusion (**left panel**), higher CFQ score indicates lower levels of cognitive defusion; psychological flexibility (**middle panel**), higher AAQ score indicates lower levels of psychological flexibility; and negative body image index (negative physical self scale, NPSS) (**right panel**), higher NPSS score indicates higher levels of negative body image concern. Not the intervention was applied only to the ACT group.

**Figure 2 ijerph-19-16519-f002:**
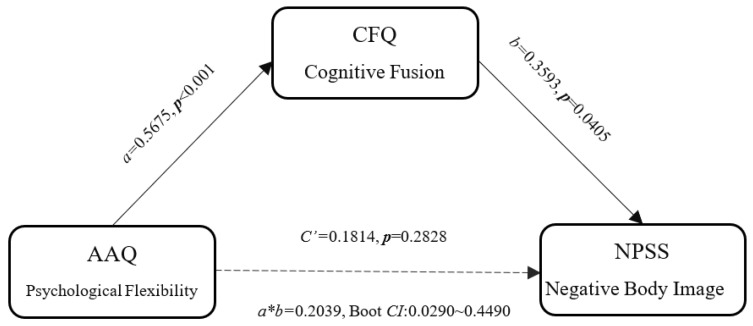
Model testing that the relation between Psychological flexibility (Post–Pre) and Negative body image attitude (Post–Pre) is mediated by Cognitive fusion (Post–Pre). The bias-corrected 95% bootstrap confidence interval (CI) for the indirect effect (a*b = 0.2039) based on bootstrap samples was above zero (0.0290~0.4490). The letters a, b, a*b, and c’ refer to estimated path coefficients. Standardized coefficients and their corresponding *p* values are presented for each path.

**Figure 3 ijerph-19-16519-f003:**
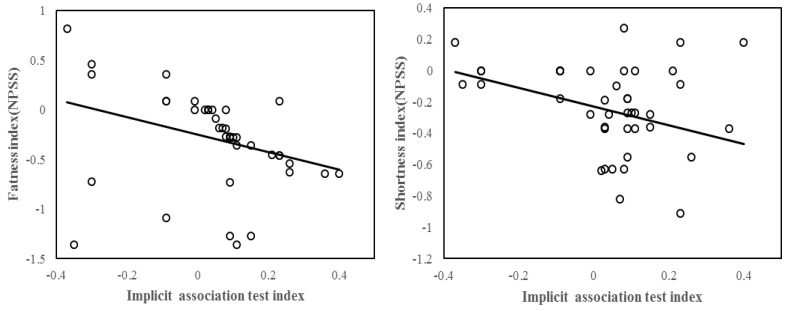
(**Left panel**), negative correlation between IAT (post−pre) and fatness index (a component of negative physical self scale). (**Right panel**), negative correlation between IAT (post−pre) and shortness index (also a component of negative physical self scale).

**Table 1 ijerph-19-16519-t001:** Participant Demographics by Condition.

Baseline Characteristics	All (*n* = 86)	ACT (*n* = 42)	Control (*n* = 44)
Age *M (SD)*	19.57 (1.32)	19.93 (0.92)	20.04 (1.18)
Gender			
male	40 (46.5%)	19 (45.2%)	21 (47.7%)
Female	46 (53.5%)	23 (54.8%)	23 (52.3%)
College Grade			
Freshmen	26 (30.2%)	13 (30.9%)	13 (29.5%)
Sophomores	21 (24.4%)	12 (28.6%)	9 (20.5%)
Junior	20 (23.2%)	10 (23.8%)	10 (22.7%)
Senior	19 (22.2%)	7 (16.7%)	12 (27.3%)

Note, ACT = the intervention group, Control = Control group.

**Table 2 ijerph-19-16519-t002:** The content of sessions based on acceptance and commitment.

Session	Brief Description
1. First meeting	Get to know each other by palying snowball game
	Introduction of the ACT therapy
	Sign the informed consent
2. Live in the present	Introduction of “contact with the present moment”
	Let participants discuss the latest social events (such as excessive weight loss)
	Mindfulness practice: mindful breathing practice; eating currant mindful practice
	Homework: Mindful breathing practice 10 min a day; “contact with the present moment”practice
3. Self-acceptance	Introduction of acceptance
	Review and share your homework
	Metaphor: “Pushing the folder”
	Acceptance of the mood practice
	Metaphor: “The devil on the ship”
	Homework: Acceptance of the mood practice; pay attention to the embarrassing events you encounter about body image in your daily life
4. Cognitive defusion	Introduce concept of separation or distance from thoughts
	Practice concept of separation or distance from thoughts
	Metaphor: Monitor
	Meditation practice: Watching your thoughts
	Carry your Keys: Create separation or distance between the self and unwanted or painful thoughts
	Presenter summariz the group intervention
	Homework: Meditation practice; think about situations in which you feel trapped
5. Self-as-context	Introduction of self as context
	Metaphor: The sky and the weather
	Practice: “You are in the process”
	Mailbox metaphor: Examine the distinction between description and evaluation of experiences; practice observing the self as the context in which experiences occur
	Homework: Mindful practice of taking self as context
6. Watch your thoughts	Pay attention to your thoughts and describe them as stories
	Singing and funny voice practice
	Lemon lemon repetition practice
	“Leaves on a stream” meditation practice
	Homework: Meditation practice; name the story
7. Values	Introduction of value
	Compass metaphor: Let undergraduates understand the difference between value and goal
	Two children in the car: Let the participants understand that value is in the present
	Imagine your 80th birthday metaphor: help undergraduates clarify the value
	Presenter summarize the group intervention
	Homework: Ask participants match their values to their own lives, to clarify the values
8. Committed action	Introduction of Commitment Action
	Review and share their homework
	Identify committed actions needed to address challenges and move in the direction of values
	Meet and resolve obstacles in action
	Sign the The Willingness and Action Plan
	Presenter summariz the group intervention
	Homework: Set a value-based goal about improving your negative body image
9. Do your best	Warm-up: Wind blowing
	Share their gain and their regrets in the process of achieving their goals
	Write a letter to the future, with the theme “face your negative body image and do your best”
	Presenter summariz the group intervention and arrange the last group intervention activity
10. Summary	Share and summarize
	Explore self-change and encourage undergraduates continue to act towards their value-base goal
	Useful resources about additional help

**Table 3 ijerph-19-16519-t003:** The procedures of IAT.

Trial Blocks	Tasks Description	Q Key	P Key
1	Attitude discrimination	Positive body image	Negative body image
2	Concept discrimination	Self	Non-self
3	Compatible combined task (excise)	Self + Positive body image	Non-self + Negative body image
4	Compatible combined task (formal)	Self + Positive body image	Non-self + Negative body image
5	Reversed concept discrimination	Non-self	Self
6	Reversed combined task (excise)	Non-self + Positive body image	Self + Negative body image
7	Reversed combined task (formal)	Non-self + Positive body image	Self + Negative body image

**Table 4 ijerph-19-16519-t004:** Descriptive data at pretreatment and posttreatment, results of the one-way ANCOVA, and effect size.

	ACT Group (*n* = 42)				Control Group (*n* = 44)				Effect Size	
	Pre		Post		Pre		Post		*F*	Cohen’s *d*
	*M*	*SD*	*M*	*SD*	*M*	*SD*	*M*	*SD*		
AAQ-II	30.80	8.44	21.07	4.53	28.07	6.89	27.53	5.88	4.18 *	0.32
CFQ-F	45.93	14.22	33.20	8.38	42.27	10.24	43.67	9.56	7.67 *	0.43
General Appearance	3.27	0.90	2.25	0.61	2.96	0.75	2.75	0.82	38.51 ***	0.96
Facial Appearance	2.52	0.65	1.74	0.53	2.18	0.55	2.23	0.56	34.99 ***	0.91
Fatness	2.44	1.09	1.79	0.77	2.33	0.80	2.41	0.90	18.93 ***	0.67
Shortness	2.66	0.86	1.81	0.77	2.19	0.74	2.33	0.70	16.45 ***	0.63
Thinness	1.82	0.84	1.33	0.39	2.19	0.59	2.25	0.50	20.32 ***	0.70

Note. AAQ-II = Acceptance and Action Questionnaire II; CFQ-F = Cognitive Fusion Questionnaire. * *p* < 0.05. *** *p* ≤ 0.001.

**Table 5 ijerph-19-16519-t005:** Predictive role of IAT (post-pre).

Variables	*β*	T	R^2^	ΔR^2^
Fatness (post-pre)				
Step1			0.01	
age	0.09	0.54		
gender	−0.01	−0.05		
Step2			0.16 *	0.15 *
age	0.12	0.78		
gender	0.02	0.14		
IAT (post-pre)	−0.39 *	−2.54		
Shortness (post-pre)				
Step1			0.01	
age	0.01	0.08		
gender	0.01	0.06		
Step2			0.13 *	0.12 *
age	0.04	0.26		
gender	0.04	0.23		
IAT (post-pre)	−0.36 *	−2.30		

Note. * *p* < 0.05.

## Data Availability

The datasets generated during and/or analyzed during the current study are available from the first author upon reasonable request.

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
