# Peer review of "Cognitive Defusion and Psychological Flexibility Predict Negative Body Image in the Chinese College Students: Evidence from Acceptance and Commitment Therapy"

_ijerph, 2022, doi:10.3390/ijerph192416519_

Round 1

Reviewer 1 Report

Dear Editor, 

The manuscript examines the effectiveness of the ACT protocol on body image in Chinese university students.

Strengths:

the literature review is adequate

statistical analyses are appropriate

experimental and control sample size is appropriate

the research design is clear enough

the psychometric instruments are adequate and validated

Weaknesses:

- In the abstract it is reported that the number of subjects who participated in the study is 92 however in the text it is reported that there are 86 subjects. Of these in the abstact 46 did the ACT sessions, in the text it is 44. Please compare.

- In the graphs comparing pre-post you show: CTQ AAQ and AAQ but in the description you refer to CFQ AAQ and NPSS , please check.

- Table four representing correlations could be taken out and summarised in words within the text.

- In the text you often use the expression "post-pre", please specify whether you refer to the "pre-post" comparison or a single score, please explain.

- The bibliography is partially dated and does not include more recent work (2020-2022).

I recommend acceptance of the article after minor revisions.

Author Response

Dear Reviewer 1,

Please find uploaded the revised manuscript entitled “Cognitive Defusion and Psychological Flexibility Predict Negative Body Image in the Chinese College Students:A RCT Trail of Acceptance and Commitment Therapy”. Below, we provide a point-by-point reply (our reply is in blue font). To help in the review process, we have marked our changes to the text in red font.

Thank you very much for your valuable and kind comments and critiques on our manuscript, which are very helpful to improve the quality of our paper. We believe that the revision has addressed all the concerns. We have added some important references and cited them appropriately. We hope that you find the current version to be ready for publication.

Sincerely yours,

Fang, et al.

Response to Reviewer

The manuscript examines the effectiveness of the ACT protocol on body image in Chinese university students.

Strengths:

the literature review is adequate

statistical analyses are appropriate

experimental and control sample size is appropriate

the research design is clear enough

the psychometric instruments are adequate and validated

Response: Thank you for your positive comments on the content of the manuscript, it gives us a lot of confidence.

Weaknesses:

- In the abstract it is reported that the number of subjects who participated in the study is 92 however in the text it is reported that there are 86 subjects. Of these in the abstact 46 did the ACT sessions, in the text it is 44. Please compare.

Response: Thank you for your valuable suggestions! We have modified the abstract accordingly.

Here, 86 young Chinese university students with high negative physical self were invited, in which 42 students received 10 sessions of group-based ACT intervention in a clinical setting while the remained acted as the control group with no intervention.

- In the graphs comparing pre-post you show: CTQ AAQ and AAQ but in the description you refer to CFQ AAQ and NPSS, please check.

Response: Thank you for your valuable suggestions! We have revised in Fig 1.

- Table four representing correlations could be taken out and summarised in words within the text.

Response: Thank you for your valuable suggestions! We have revised:

Cognitive defusion during intervention Is psychological flexibility enhancement the underlying basis for negative body image improvement? What is the role of cognitive de-fusion then?  Further analyses revealed a significant correlation between cognitive fusion(post) and negative body image scores(post), r=.464, p=.002. Psychological flexibility(post) and cognitive defusion have(post) the strongest correlation, r=.568, p<.001, Significant positive correlation between psychological flexibility(post) and negative body image(post), r=.389, p=.011.

- In the text you often use the expression "post-pre", please specify whether you refer to the "pre-post" comparison or a single score, please explain.

Response: Thank you for your valuable suggestions! "post-pre" means refers to the pre-test score minus the post test score.

- The bibliography is partially dated and does not include more recent work (2020-2022).

Response: Thank you for your valuable suggestions! We have added some important and recent references and cited them appropriately. They are as follows:

  • Alharballeh, S., & Dodeen, H. (2021). Prevalence of body image dissatisfaction among youth in the United Arab Emirates: gender, age, and body mass index differences. Current Psychology. https://doi.org/10.1007/s12144-021-01551-8
  • Barney, J. L., Barrett, T. S., Lensegrav-Benson, T., Quakenbush, B., & Twohig, M. P. (2022). Examining a mediation model of body image-related cognitive fusion, intuitive eating, and eating disorder symptom severity in a clinical sample. Eating and Weight Disorders - Studies on Anorexia, Bulimia and Obesity, 27(6), 2181-2192. https://doi.org/10.1007/s40519-021-01352-9
  • Bennett, B. L., Wagner, A. F., & Latner, J. D. (2022). Body Checking and Body Image Avoidance as Partial Mediators of the Relationship between Internalized Weight Bias and Body Dissatisfaction. International Journal of Environmental Research and Public Health, 19(16), 9785. https://www.mdpi.com/1660-4601/19/16/9785
  • Fang, S., & Ding, D. (2022). Which outcome variables are associated with psychological inflexibility/flexibility for chronic pain patients? A three level meta-analysis. Frontiers in Psychology. 13:1069748. https://doi.org/3389/fpsyg.2022.1069748
  • Fang, S., & Wang, W. (2011). Act in practice: Case Conceptualization in Acceptance and Commitment Therapy. Chongqing: Chongqing University Press.
  • Fogelkvist, M., Gustafsson, S. A., Kjellin, L., & Parling, T. (2020). Acceptance and commitment therapy to reduce eating disorder symptoms and body image problems in patients with residual eating disorder symptoms: A randomized controlled trial. Body Image, 32, 155-166. https://doi.org/10.1016/j.bodyim.2020.01.002
  • Moradi, F., Ghadiri-Anari, A., Dehghani, A., Reza Vaziri, S., & Enjezab, B. (2020). The effectiveness of counseling based on acceptance and commitment therapy on body image and self-esteem in polycystic ovary syndrome: An RCT. International Journal of Reproductive Biomedicine, 18(4), 243-252. https://doi.org/10.18502/ijrm.v13i4.6887
  • Romano, K. A., Heron, K. E., & Ebener, D. (2021). Associations among weight suppression, self-acceptance, negative body image, and eating disorder behaviors among women with eating disorder symptoms. Women & Health, 61(8), 791-799. https://doi.org/10.1080/03630242.2021.1970082
  • Selvi, K., Parling, T., Ljótsson, B., Welch, E., & Ghaderi, A. (2021). Two randomized controlled trials of the efficacy of acceptance and commitment therapy-based educational course for body shape dissatisfaction. Scandinavian Journal of Psychology, 62(2), 249-258. https://doi.org/10.1111/sjop.12684
  • Shepherd, L., Turner, A., Reynolds, D. P., & Thompson, A. R. (2020). Acceptance and commitment therapy for appearance anxiety: three case studies. Scars, Burns & Healing, 6, 2059513120967584. https://doi.org/10.1177/2059513120967584
  • Stojcic, I., Dong, X., & Ren, X. (2020). Body Image and Sociocultural Predictors of Body Image Dissatisfaction in Croatian and Chinese Women. Frontiers in Psychology, 11. https://doi.org/10.3389/fpsyg.2020.00731
  • Wallner, C., Kruber, S., Adebayo, S. O., Ayandele, O., Namatame, H., Olonisakin, T. T., O. Olapegba, P., Sawamiya, Y., Suzuki, T., Yamamiya, Y., Wagner, M. J., Drysch, M., Lehnhardt, M., & Behr, B. (2022). Interethnic Influencing Factors Regarding Buttocks Body Image in Women from Nigeria, Germany, USA and Japan. International Journal of Environmental Research and Public Health, 19(20), 13212. https://www.mdpi.com/1660-4601/19/20/13212
  • Wang, J., & Fang, S. (2022). Effects of Internet-Based Acceptance and Commitment Therapy (IACT) on Adolescents: A Systematic Review and Meta-Analysis. International Journal of Mental Health Promotion, https://doi.org/10.32604/ijmhp.2023.025304
  • Zucchelli, F., White, P., & Williamson, H. (2020). Experiential avoidance and cognitive fusion mediate the relationship between body evaluation and unhelpful body image coping strategies in individuals with visible differences. Body Image, 32, 121-127. https://doi.org/https://doi.org/10.1016/j.bodyim.2019.12.002

I recommend acceptance of the article after minor revisions.

Thank you so much for your acknowledgment!

Reviewer 2 Report

As I wrote in my review, the paper addresses universal problem of perception and attitudes to one’s body. it is universal, as in all times people tried to fit some ideal image that existed or exist now.

Authors offered relevant literature review, particular strength of this review was accent on Eastern and particularly Chinese specifics of body image. In modern world it is very important to study and emphasize the specifics of basic mental processes from non-Western countries.

They also presented clear and detailed description of all procedures and design that makes it easy to understand and evaluate their results. Results also presented in much details, with relevant statistics and support hypothesis and conclusions.

My main concern, very important, that should be obligatory corrected is title: all the paper is very consistent interms of background, design and results, but the whole paper is about young Chinese adults, not total adult population as it is stated in the title. I believe, if they will clarify this moment in the title, the paper can be accepted as it is.

Author Response

Dear Reviewer 2,

As I wrote in my review, the paper addresses universal problem of perception and attitudes to one’s body. it is universal, as in all times people tried to fit some ideal image that existed or exist now.

Authors offered relevant literature review, particular strength of this review was accent on Eastern and particularly Chinese specifics of body image. In modern world it is very important to study and emphasize the specifics of basic mental processes from non-Western countries.

They also presented clear and detailed description of all procedures and design that makes it easy to understand and evaluate their results. Results also presented in much details, with relevant statistics and support hypothesis and conclusions.

Response: Thank you so much for your acknowledgment!

My main concern, very important, that should be obligatory corrected is title: all the paper is very consistent in terms of background, design and results, but the whole paper is about young Chinese adults, not total adult population as it is stated in the title. I believe, if they will clarify this moment in the title, the paper can be accepted as it is.

Response: Thank you for your valuable suggestions! We have revised the title of the manuscript and the corresponding statements in the text.

Cognitive Defusion and Psychological Flexibility Predict Negative Body Image in the Chinese College Students:A RCT Trail of Acceptance and Commitment Therapy

Reviewer 3 Report

I ask the authors to make the following corrections/completions:

1.The abstract must be redone. It must have the classic structure and it should have 200-250 words. Indicate the type of study in the abstract.

2. At the end of the introduction, immediately after the aim, the objectives of the study and the prespecified hypotheses must be stated.

3. At 2.1 Study type and study design.

Outline the key elements of the study type.

Describe where the study was done and in what period. Without "XX University". If the study was done legally, it is assumed by the respective university.

At 2.2 Participants

Indicate the eligibility criteria for the group of cases. Indicate the rejection criteria for the group of cases.

Explain the reasons for the choice of cases and controls.

In 2.4, a sentence must be added to clearly state which variables are studied (for example, independent variables: age and gender; dependent variables: psychological flexibility, cognitive defusion, negative body image, etc.).

R287-R288 refer to the methodology and should be brought to 2.4 in the phrase with the variables.

At 2.5 Ethical consideration

The phrase from R146-150 must be added.

You must specify the number of endorsement/approval from the Research Ethics Commission issued by "University XX".

4. At Results

The general characteristics of the study participants (eg, demographics, etc.) should be presented in a table (Table I).

A summary table must be made for the main results.

All figures must have values ​​(with value shown).

5. At Conclusions

The article must have a distinct section for conclusions. These must be derived from the research results.

6. At Limits of the study

A sentence must be written about the limits of the study.

7. Authors contribution???

8. The bibliography must be written according to MDPI recommendations. An example was given at the end, in the IJERPH template used by you for this paper, it just had to be respected.

Thank you for your work!

I hope that the suggestions made by me will help you to improve the quality of the paper.

Author Response

Reply to Reviewer 3:

Thank you very much for your valuable and kind comments and critiques on our manuscript, which are very helpful to improve the quality of our paper. A thoroughly revised version of our manuscript has been made.

Please review uploaded the revised manuscript entitled “Cognitive Defusion and Psychological Flexibility Predict Negative Body Image in the Chinese College Students: Evidence from Acceptance and Commitment Therapy”. Below, we provide a point-by-point reply (our reply is in blue font). To help in the review process, we have marked our changes to the text in red font.

Sincerely yours,

Fang, et al.

1.The abstract must be redone. It must have the classic structure and it should have 200-250 words. Indicate the type of study in the abstract.

Response: Thanks for your valuable advice!

We added the type of study in the abstract of the manuscript.

The study was 2 (condition: intervention vs waitlist control) × 2 (time: pre- and post-test) mixed design.

  1. At the end of the introduction, immediately after the aim, the objectives of the study and the prespecified hypotheses must be stated.

Response: Thanks for your valuable advice!

We added the research objectives of this study:

In summary, the main purpose of this study is to explore whether group-based acceptance and commitment therapy can reduce negative body image among college students and what are the intervention mechanisms involved?

  1. At 2.1 Study type and study design.

Outline the key elements of the study type.

Describe where the study was done and in what period. Without "XX University". If the study was done legally, it is assumed by the respective university.

Response: Thank you for your valuable advice!

It is clearly written in the main text that:

The study was 2 (condition: intervention vs waitlist control) × 2 (time: pre- and post-test) mixed design, with a final sample of 86 students (ACT intervention group: n = 42; waitlist control group: n = 44). G*Power 3.1 showed that the statistical power is 0.94 when linear regression was used with sample size of 86, default effect size (R2) of 0.15, and significant level of 0.05.

This study was conducted at Anhui Normal University. For the purpose of blind review, the previous manuscript was written "XX University", but now we explicitly write Anhui Normal University.

At 2.2 Participants

Indicate the eligibility criteria for the group of cases. Indicate the rejection criteria for the group of cases.

Response: Thank you for your valuable advice!

The inclusion criteria of intervention group were being aged 18 years or older and self-reporting average score on each dimension of negative body image greater than 3. Exclusion criteria assessed via self-report measures and one-to-one online semi-structured interviews with a trained research assistant were suicidal intentions, use of psychiatric medicines, receiving psychological counseling or other treatment, and not being willing to undergo the ACT intervention.

Explain the reasons for the choice of cases and controls.

Response: Thank you for your valuable advice!

The choice of cases and controls is to test whether group-based acceptance and commitment therapy can reduce negative body image in college students and to explore the intervention mechanisms involved.

In 2.4, a sentence must be added to clearly state which variables are studied (for example, independent variables: age and gender; dependent variables: psychological flexibility, cognitive defusion, negative body image, etc.).

Response: Thank you for your valuable advice!

To test the models of cognitive defusion and psychological flexibility relate to negative body image,

This sentence in 2.4 already indicates what are the dependent variables to be explored.

R287-R288 refer to the methodology and should be brought to 2.4 in the phrase with the variables.

Response: Thank you for your valuable advice!

Modified in the main text.

At 2.5 Ethical consideration

The phrase from R146-150 must be added.

You must specify the number of endorsement/approval from the Research Ethics Commission issued by "University XX".

Response: Thank you for your valuable advice!

Ethical approval number has been added to the text:

This study was approved by the Ethical Committee of Anhui Normal University (approval number: 2018076).

  1. At Results

The general characteristics of the study participants (eg, demographics, etc.) should be presented in a table (Table I).

All figures must have values ​​(with value shown).

Response: Thank you for your valuable advice!

We have added Table 1 as follows:

Table 1. Participant Demographics by Condition

Baseline characteristics

All (n = 86)

ACT (n = 42)

Control (n = 44)

Age M (SD)

19.57(1.32)

19.93(0.92)

20.04(1.18)

Gender

male

40(46.5%)

19(45.2%)

21(47.7%)

Female

46(53.5%)

23(54.8%)

23(52.3%)

College Grade

Freshmen

26(30.2%)

13(30.9%)

13(29.5%)

Sophomores

21(24.4%)

12(28.6%)

9(20.5%)

Junior

20(23.2%)

10(23.8%)

10(22.7%)

Senior

19(22.2%)

7(16.7%)

12(27.3%)

Note, ACT = the intervention group, Control = Control group

After checking, all the figures have values now.

  1. At Conclusions

The article must have a distinct section for conclusions. These must be derived from the research results.

Response: Thank you for your valuable advice!

The conclusion of the article has been expounded in the discussion section.

  1. At Limits of the study

A sentence must be written about the limits of the study.

Response: Thank you for your valuable advice!

Already added:

This study did not conduct follow-up measurements to examine the persistence of the intervention study.

  1. Authors contribution???

Response: Thank you for your valuable advice!

Already added:

Author Contributions: SF, KH contributed to conceptualization, methodology, visualization, writing–review & editing. SF, PJ, and MH designed and implemented this experiment, organized the database, performed the statistical analysis. SF, DD, and KH contributed to formal analysis and wrote original draft. SF contributed to supervision, project administration, funding acquisition. All Authors contributed to the article and approved the submitted version.

  1. The bibliography must be written according to MDPI recommendations. An example was given at the end, in the IJERPH template used by you for this paper, it just had to be respected.

Response: Thank you for your valuable advice!

We checked every reference; all references are relevant to the contents of the manuscript and correspond to the citations in the text. 

Reviewer 4 Report

Thank you very much for allowing me to read this interesting manuscript. I appreciate the interest of researchers in trying to investigate if the ACT is effective in intervening with the negative body image. Nevertheless, several improvements are required to the current iteration before publication can be recommended. Namely, I would suggest clarifying many aspects of the methodology. Please see the below comments and recommendations:

· The manuscript is consider as “RCT”, have you consider any checklist in order to evaluate the quality? Have you consider consort statement?

· Previously, the study protocol has been recorded, not only a bibliographic reference to the protocol for the application to the intervention group?

· The control group has not received any intervention, why was this design decided in an RCT?

·  The study is using a convenience sampling method, only the control group appears to be randomised, and it  allocation is not clear.

·  Why the control group was paid and not the experimental group? Could this affect the results of your study?

· Inclusion/exclusion criteria are not clearly reported.

· There is no mention of whether there is any form of blinding.

· The socio-demographic characteristics of the samples are not presented. It is not detailed whether homogeneity exists.

· It is not reported if all persons in the intervention group complete 10 sessions.

·   Lines 426-427 “The findings from the present study should be interpreted with caution, given the limitations of the study”, could explain the limitations?

Based on the above,  I regret to state that I cannot recommend publication at this time. 

Author Response

Reply to Reviewer 4:

Thank you very much for your valuable and kind comments and critiques on our manuscript, which are very helpful to improve the quality of our paper. A thoroughly revised version of our manuscript has been made.

Please review uploaded the revised manuscript entitled “Cognitive Defusion and Psychological Flexibility Predict Negative Body Image in the Chinese College Students:Evidence from Acceptance and Commitment Therapy”. Below, we provide a point-by-point reply (our reply is in blue font). To help in the review process, we have marked our changes to the text in red font.

Sincerely yours,

Fang, et al.

  • The manuscript is consider as “RCT”, have you consider any checklist in order to evaluate the quality? Have you consider consort statement?

Response: Thank you for your valuable suggestions!

As you pointed out, it's true that our study was not a strictly randomized controlled study. Due to our negligence in writing, we put a wrong expression in the title of the manuscript. Now we have changed the title of the manuscript. There is no reference in the text to the fact that the study is a randomized controlled study.

Cognitive Defusion and Psychological Flexibility Predict Negative Body Image in the Chinese College Students: Evidence from Acceptance and Commitment Therapy

  • Previously, the study protocol has been recorded, not only a bibliographic reference to the protocol for the application to the intervention group?

Response: Thank you for your valuable suggestions!

The study protocol of the intervention group was specially designed for college students with negative body image according to the principles and techniques of acceptance and commitment therapy, and the content validity was verified by acceptance and commitment therapy consultants and supervisors.

  • The control group has not received any intervention, why was this design decided in an RCT?

Response: Thanks for your valuable advice!

This is a mistake in the description of the title of our manuscript. We have modified the title of the manuscript.

  • The study is using a convenience sampling method, only the control group appears to be randomised, and it  allocation is not clear.

Response: Thank you for your valuable advice!

We have amended the statement in the text as follows:

The study was conducted at the first author's university.  Using a convenience sampling method, we contacted university counselors and distributed a total of 1,000 questionnaires. 968 questionnaires were returned(96.8% effective rate). We identified 463 of the 968 surveys with an average score on each dimension of negative body image greater than 3. Among them, there were 244 males and 219 females, aged between 17 and 22. Participants in this study were recruited from these 463 university students. The inclusion criteria of intervention group were being aged 18 years or older and self-reporting average score on each dimension of negative body image greater than 3. Exclusion criteria assessed via self-report measures and one-to-one online semi-structured interviews with a trained research assistant were suicidal intentions, use of psychiatric medicines, receiving psychological counseling or other treatment, and not being willing to undergo the ACT intervention. 46 participants, including 22 males and 24 females, were recruited as the intervention group. 4 participants withdrew from intervention due to their private reasons in the later stage of the process. In the end, there were 42 participants in the intervention group, including 19 males and 23 females, aged from 18 to 22. At the same time, 46 out of the remaining students were randomly selected as the control group(2 participants withdrew for their private reasons), Finally, the control group consisted of 44 participants, including 21 males and 23 females, aged from 18 to 22, during which no intervention was performed. Statistical power of the sample size was calculated using G*Power 3.1 (Faul, Erdfelder, Buchner, & Lang, 2009). The statistical power for a sample size of 86 is 0.96 when ANCOVA was used with group size of four, effect size (f) of 0.4, and a significance level of 0.05 (Mandavia, et al., 2015).

  • Why the control group was paid and not the experimental group? Could this affect the results of your study?

Response: Thanks for your valuable advice!

It was our misrepresentation that all 86 participants were actually paid a certain amount at the end of the experiment.

  • Inclusion/exclusion criteria are not clearly reported.

Response: Thanks for your valuable advice!

Our supplementary report is as follows:

Participants in this study were recruited from these 463 university students. The inclusion criteria of intervention group were being aged 18 years or older and self-reporting average score on each dimension of negative body image greater than 3. Exclusion criteria assessed via self-report measures and one-to-one online semi-structured interviews with a trained research assistant were suicidal intentions, use of psychiatric medicines, receiving psychological counseling or other treatment, and not being willing to undergo the ACT intervention.

  • There is no mention of whether there is any form of blinding.

Response: Thanks for your valuable advice!

Since this study is not a strictly randomized controlled study, there is no mention of any form of blindness.

  • The socio-demographic characteristics of the samples are not presented. It is not detailed whether homogeneity exists.

Response: Thanks for your valuable advice!

We add the socio-demographic characteristics of the samples, and in the results section state that the two groups are homogeneous in the pre-test.

  • It is not reported if all persons in the intervention group complete 10 sessions.

Response: Thanks for your valuable advice!

At the beginning, 46 university students were recruited to participate in the intervention group experiment, four did not complete the 10 interventions and eventually all 42 completed the 10 interventions and took the post-test.

  • Lines 426-427 “The findings from the present study should be interpreted with caution, given the limitations of the study”, could explain the limitations?

Response: Thanks for your valuable advice!

For the explanation of limitations, we have added the following explanation in addition to the original one: This study did not conduct follow-up measurements to examine the persistence of the intervention study. In addition, the fact that this study was not a strictly randomized controlled trial is also a limitation of this study.

Round 2

Reviewer 3 Report

I ask the authors to make the following changes/corrections:

1. Bibliographic references must be placed in the order of their entry in the text in square brackets, for example: [1].

2. The number and date of approval of the study by the University's Research Ethics Commission must be written.

3. The study must have CONCLUSIONS.

4. All bibliographic references must be written according to MDPI requirements.

E.g.:

Alharballeh, S; Dodeen, H. Prevalence of body image dissatisfaction among youth in the United Arab Emirates: gender, age, and body mass index differences. Curr Psychol. 2021 Mar, 1: 1-10. Doi: 10.1007/s12144-021-01551-8. PMID: 33679115; PMCID: PMC7919234.

Thanks for the work done!

Author Response

Reply to Reviewer 3:

  1. Bibliographic references must be placed in the order of their entry in the text in square brackets, for example: [1].

Response: Thanks for your valuable advice!

We checked every reference; all references are relevant to the contents of the manuscript and correspond to the citations in the text. We cited and collated references in accordance with APA format. Once the manuscript is accepted, we will strictly follow the "Bibliographic references must be placed in the order of their entry in the text in square brackets".

  1. The number and date of approval of the study by the University's Research Ethics Commission must be written.

Response: Thanks for your valuable advice!

The number and date of approval of the study by the university ethics committee has been written.

This study was approved by the Ethical Committee of Anhui Normal University (approval number: 2018076, October 10, 2018).

  1. The study must have CONCLUSIONS.

Response: Thank you for your valuable comments!

We have stated the conclusions of this study very clearly in the first paragraph of the Discussion section. In addition, we have outlined the main findings of this study inside the abstract.

We showed that (1) cognitive fusion correlated with psychological flexibility, and each well predicted body dissatisfaction in the higher negative body image sample; (2) Cognitive defusion predicted negative body image improvement. Psychological flexibility was enhanced via ACT intervention, yet its influence on body image attitude was mediated by cognitive defusion; (3) ACT intervention modulated the implicit processing of fatness and thinness, which are special in Chinese culture.

  1. All bibliographic references must be written according to MDPI requirements.

E.g.:

Alharballeh, S; Dodeen, H. Prevalence of body image dissatisfaction among youth in the United Arab Emirates: gender, age, and body mass index differences. Curr Psychol. 2021 Mar, 1: 1-10. Doi: 10.1007/s12144-021-01551-8. PMID: 33679115; PMCID: PMC7919234.

Response: Thank you for your valuable comments!

After the manuscript is accepted, we will strictly follow the MDPI requirements to organize the references.

Reviewer 4 Report

The authors provided satisfactory revisions to the article. Thank you for responding so comprehensively to my comments and for clarifying that it is not an RCT.

The limitations of the study are again not very precise, please try to improve them.

Author Response

Reply to Reviewer 4:

The authors provided satisfactory revisions to the article. Thank you for responding so comprehensively to my comments and for clarifying that it is not an RCT.

Response: Thank you so much for your acknowledgment!

The limitations of the study are again not very precise, please try to improve them.

Response: Thank you for your valuable suggestions!

We supplemented and refined the limitations of the study.

The present study measured psychological flexibility using the Acceptance and Action Questionnaire II (AAQ-II), which measures only one dimension of psychological flexibility. In addition, this study did not have follow-up measurements to examine the durability of the intervention study effects. Given these limitations, the findings from this study should be interpreted with caution.
